# Renal Expression and Localization of the Receptor for (Pro)renin and Its Ligands in Rodent Models of Diabetes, Metabolic Syndrome, and Age-Dependent Focal and Segmental Glomerulosclerosis

**DOI:** 10.3390/ijms25042217

**Published:** 2024-02-12

**Authors:** Carla Iacobini, Martina Vitale, Federica Sentinelli, Jonida Haxhi, Giuseppe Pugliese, Stefano Menini

**Affiliations:** 1Department of Clinical and Molecular Medicine, “La Sapienza” University, 00189 Rome, Italy; carla.iacobini@uniroma1.it (C.I.); martina.vitale@uniroma1.it (M.V.); jonida.haxhi@uniroma1.it (J.H.); stefano.menini@uniroma1.it (S.M.); 2Department of Public Health and Infectious Diseases, “La Sapienza” University, 00189 Rome, Italy; federica.sentinelli@uniroma1.it

**Keywords:** cyclooxygenase 2, diabetic nephropathy, hyperglycemia, Milan normotensive rats, soluble prorenin receptor, podocytes

## Abstract

The (pro)renin receptor ((P)RR), a versatile protein found in various organs, including the kidney, is implicated in cardiometabolic conditions like diabetes, hypertension, and dyslipidemia, potentially contributing to organ damage. Importantly, changes in (pro)renin/(P)RR system localization during renal injury, a critical information base, remain unexplored. This study investigates the expression and topographic localization of the full length (FL)-(P)RR, its ligands (renin and prorenin), and its target cyclooxygenase-2 and found that they are upregulated in three distinct animal models of renal injury. The protein expression of these targets, initially confined to specific tubular renal cell types in control animals, increases in renal injury models, extending to glomerular cells. (P)RR gene expression correlates with protein changes in a genetic model of focal and segmental glomerulosclerosis. However, in diabetic and high-fat-fed mice, (P)RR mRNA levels contradict FL-(P)RR immunoreactivity. Research on diabetic mice kidneys and human podocytes exposed to diabetic glucose levels suggests that this inconsistency may result from disrupted intracellular (P)RR processing, likely due to increased Munc18-1 interacting protein 3. It follows that changes in FL-(P)RR cellular content mechanisms are specific to renal disease etiology, emphasizing the need for consideration in future studies exploring this receptor’s involvement in renal damage of different origins.

## 1. Introduction

The (Pro)renin receptor, (P)RR for short, is characterized by a singular transmembrane protein capable of binding renin and prorenin (the proenzyme precursor of renin) with the same affinity [1]. Research on (P)RR suggests that its increased expression contributes to cardiometabolic diseases, including diabetes, hypertension, and dyslipidemia, and associated organ damage [2].

The activity of (P)RR has been associated with the non-proteolytic activation of prorenin, leading to localized activation of the renin angiotensin system (RAS) and initiation of RAS-dependent and independent signaling pathways [3]. In fact, (P)RR activation can directly upregulate profibrotic gene expression via activation of the mitogen-activated protein kinase–extracellular signal-regulated kinase 1/2 pathway and stimulate the expression of pro-inflammatory genes through modulation of cyclooxygenase-2 (COX-2), a key mediator of inflammatory pathways [4]. As a result, the binding of (pro)renin to (P)RR can trigger pro-inflammatory and fibrogenic responses regardless of angiotensin II generation. This indicates an autonomous and additional function of (P)RR in the progression of renal injury, especially in diabetic nephropathy (DN). In fact, elevated plasma prorenin levels have been observed in diabetic patients, correlating with DN and sometimes occurring before the onset of microalbuminuria [5,6,7,8]. All together, these findings have spurred interest in the potential involvement of (P)RR in diabetic kidney disease, but also in renal injury stemming from other cardiometabolic abnormalities [2]. In DN, changes in renal levels of full-length (FL)-(P)RR and circulating levels of its soluble form [s(P)RR] have been suggested to play a pathogenic role. However, due to inconsistent findings [9,10,11,12], the in vivo pathological significance of the (P)RR system, especially as a component of RAS, remains a topic of ongoing debate.

This debate aside, the fundamental mechanisms that regulate (P)RR expression and post-translational processing remain inadequately understood. More importantly, the alterations in tissue/cellular localization of this receptor during renal injury, and their consistency across renal diseases of various origins, are not well elucidated.

The aim of this study is to analyze the expression and topographic localization of FL-(P)RR, along with its ligands renin and prorenin, and its target COX-2 [4], in the kidneys of two mouse models of metabolic disorders (diabetes and metabolic syndrome) and a rat model of classic focal segmental glomerulosclerosis (FSGS), with the rat being of the Milan normotensive strain (MNS) [13,14,15]. Due to inconsistencies between the gene expression of (P)RR and protein levels of FL-(P)RR in the kidneys of the mouse models of metabolic disorders, especially in diabetic (Diab) mice, we conducted an investigation into the expression of enzymatic and adapter proteins involved in the intracellular processing of FL-(P)RR in both diabetic kidneys and podocytes cultured in high glucose (HG) conditions. Finally, we analyzed the impact of these expression changes on FL-(P)RR cell distribution and the secretion of s(P)RR by podocytes.

## 2. Results

### 2.1. Kidney Specimens and Renal Structural and Functional Features of the Experimental Rodent Models

The kidney specimens analyzed in this study were obtained from our biobank of formalin fixed and paraffin embedded (FFPE) tissues sourced from rodent models utilized in previous studies conducted by our research group. The kidney tissue from three experimental models of renal damage were analyzed: (1) 129/Sv mice rendered diabetic with streptozotocin and coeval non diabetic controls, followed for 16 weeks [16]; (2) C57BL/6J mice fed a high fat diet containing 60% of total calories from fat for 16 weeks, and coeval controls fed a normal fat diet (NFD) containing 10% of total calories from fat [17]; (3) MNS rats aged two and nine months, which develop age-dependent focal and segmental glomerulosclerosis starting from the third/fourth month of life [18]. Progenitor Wistar (WS) rats of the same age were used as controls [14]. In MNS rats, an upregulated synthesis of thromboxane A2 within the glomeruli, which is known to be orchestrated by the sequential action of COX enzymes [19], is a significant factor in the progression of kidney disease [20].

As previously shown [14,16,17], all three experimental models exhibit renal structural and functional abnormalities, albeit to varying degrees (Table 1).

Diabetic (Diab) and HFD-fed mice (HFD) presented with diffuse glomerulosclerosis with minimal focal glomerulosclerosis and a lack of obvious tubular damage. The glomerular sclerosis index score (GSI) was significantly increased in both Diab and HFD-fed mice compared to their respective controls, but Diab mice showed twice the glomerulosclerosis score of HFD mice. Similarly, total proteinuria was increased in both groups, but Diab mice showed values more than double those of HFD mice [16,17]. There were no differences between the two strains of control mice (129/Sv and C57BL/6J) for any structural or functional parameter [16,17]. At two months of age, no structural or functional damage was evident in the MNS rats as compared with age-matched WS control rats. However, by the age of nine months, renal lesions in MNS rats were notably more pronounced compared to mouse models. This included extensive segmental or global glomerulosclerosis in approximately half of the glomeruli, along with widespread interstitial fibrosis and tubular damage. Consistently, proteinuria was increased by approximately 30 times in MNS rats compared to WS rats [14], which is several-fold higher than in Diab and HFD mice [16,17].

No significant differences in the basal expression levels of the various targets under investigation were observed between the control groups of mice (C57BL/6J and 129/Sv). Therefore, to simplify the presentation of results and avoid redundancies, only data from C57BL/6J mice will be shown as control (Ctr) for both the Diab and HFD groups.

### 2.2. Immunostaining for FL-(P)RR

In Ctr mice, positive staining for FL-(P)RR is predominantly limited to the collecting duct and a few cells in the distal nephron segments near the glomeruli. In contrast, cortical staining for FL-(P)RR is increased in HFD and, particularly, Diab mice (Figure 1A), with a diffuse and intense positivity observed in cells of the distal convoluted tubules (DCT) (Figure 1A,B).

Strong positive staining also appeared in glomerular cells of Diab and, to a lesser extent, HFD mice, particularly in cells exhibiting the morphological features and topographical localization characteristic of podocytes, and in endothelial cells lining the walls of small kidney arterioles (Figure 1B). Generally, the pattern of positivity for FL-(P)RR was mainly cytoplasmic/membranous, although occasionally strong perinuclear positivity was also noted. In tubular epithelial cells and collecting duct cells, it almost exclusively involved the apical membrane (Figure 1B).

In WS rats, the staining pattern for FL-(P)RR was similar to that observed in CTR mice, both at two and nine months of age. In contrast, MNS rats showed increased cortical staining for FL-(P)RR compared to WS controls at two months of age, before any signs of renal damage could be observed [14,18]. This staining became more pronounced at nine months, especially at the glomerular level, primarily affecting podocytes, although other glomerular cell types were not spared (Figure 1C).

### 2.3. Immunostaining for (Pro)renin (Total Renin)

We subsequently conducted IHC for renin using a polyclonal antibody that specifically recognizes mature renin. We optimized a standard protocol involving antigen retrieval, followed by primary and secondary antibody reactions, to achieve robust and selective detection of juxtaglomerular cells. When employing this protocol, we observed no significant differences between the rodent groups, except for a minor, non-significant increase in HFD mice (Figure 2A).

Since the antibody used does not recognize prorenin, we hypothesized that it might also not recognize non-proteolytically activated prorenin that originates from binding to (P)RR. Consequently, we conducted renin IHC on sequential sections of the same kidney blocks after trypsin activation of prorenin to detect total renin (i.e., renin and prorenin), mirroring the method employed for determining overall plasma renin activity [21]. Pretreatment with trypsin resulted in a significant increase in cortical staining in Diab mice and, to a lesser extent, in HFD mice, while no additional positivity was observed in Ctr mice (Figure 2B). In addition, IHC staining for FL-(P)RR on sequential sections revealed a virtually identical pattern of staining for FL-(P)RR and total renin at both tubular and glomerular levels (Figure 2C). Total renin immunostaining was also increased in MNS rats compared to WS controls as early as two months of age, exhibiting a pattern identical to that observed in Diab mice (Figure 2D). In contrast to WS controls, where positive staining was limited to the juxtaglomerular cells, total renin positivity was observed in distal nephron segments and glomerular cells of MNS rats, particularly at nine months of age.

### 2.4. Immunostaining for COX-2

As the binding of (P)RR upregulates COX-2 in kidney tissue [4,22], and glucose promotes COX-2 expression via (P)RR activity [23], we assessed COX-2 protein content via IHC. In Ctr mice, COX-2 immunostaining was limited to the cells lining the wall of the distal tubule where it contacts the glomerulus, corresponding to the region of the macula densa (Figure 3A).

In contrast, a pattern of increased immunostaining, also involving the glomeruli and similar to that of FL-(P)RR, was observed in both Diab and HFD mice, with the former showing the greatest increase. Similar to the findings in mice, COX-2 immunostaining was also higher in MNS rats at two and, particularly, nine months of age, compared to the age-matched WS rats (Figure 3B).

### 2.5. mRNA Levels of (P)RR and Factors Involved in Its Intracellular Processing

The mRNA levels of PRR in HFD and, particularly, Diab mice showed opposite behavior with respect to immunoreactivity, being reduced compared to Ctr mice. Conversely, the gene expression of (P)RR was consistent with protein findings in MNS rats, as it was significantly upregulated in both age groups compared to coeval WS rats (Figure 4A).

To investigate the potential mechanism contributing to the disparity between mRNA and protein changes in Diab and HFD mice, as well as between the mouse models and the rodent model of renal damage, we analyzed the mRNA expression of site-1 protease (S1P) and furin, the primary proteases involved in the intracellular cleavage of FL-PRR and the formation of sPRR [24,25]. RT-PCR analysis did not reveal any differences in gene expression of the proteases in any of the three rodent models (Figure 4B,C). We also assessed the gene expression of Munc18-1 interacting protein 3 (Mint3), an adapter protein involved in the regulation of furin activity [26], and found increased mRNA levels in Diab mice and, to a lesser extent, HFD mice, but not in MNS rats, compared to the respective controls (Figure 4D).

### 2.6. Impact of Elevated Glucose Concentration on (P)RR Expression and Intracellular Processing in Podocytes

HG induced a change in the staining pattern of FL-(P)RR immunofluorescence in podocytes, shifting from primarily perinuclear localization to a more diffuse cytoplasmic immunostaining (Figure 5A).

If performed without cell membrane permeabilization using Triton X-100 pretreatment, the immunostaining revealed accentuation on the plasma membrane in podocytes treated with HG (Figure 5B). In contrast, (P)RR gene expression was decreased in podocytes cultured in HG conditions compared to podocytes cultured in normal glucose (NG) conditions (Figure 5C). As observed in the kidneys of Diab mice, HG levels did not modify the mRNA levels of the proteases furin and S1P, but induced an upregulation of the adapter protein Mint3 (Figure 5D–F). In parallel with these molecular changes and the modified cellular distribution of FL-(P)RR, HG induced a reduction in the formation and secretion of soluble (s)PRR. This was evidenced by a notable decrease in its levels within the culture medium (Figure 5G).

## 3. Discussion

This study illustrates alterations in the expression and topographic localization of (P)RR in the kidneys of three distinct animal models of renal injury. In mice exhibiting renal damage linked to glucose and/or lipid dysmetabolism, and in rats with age-related FSGS, the rise in positivity for FL-(P)RR is concomitant with an increase in its ligands, predominantly prorenin, and one of the primary targets of the FL-(P)RR signaling pathway, the inducible enzyme COX-2 [4,22]. The expression of FL-(P)RR, (pro)renin, and COX-2, which is limited to a specific type of specialized cells in the nephron and its vasculature in control animals, involves a broader range of cells in the tubular and glomerular compartments, including podocytes, in the damaged kidneys of all rodent models. As an accessory subunit of the vacuolar H+-ATPase, (P)RR is essential for maintaining normal podocyte structure and function [27]. However, its upregulation in podocytes has been suggested to play a role in the pathogenesis of diabetic glomerulopathy and IgA nephropathy, impacting their structure and function through the activation of various intracellular signaling pathways [3,28,29,30]. In addition, recent studies have demonstrated that the binding of recombinant (pro)renin to the (P)RR increases profibrotic factors through a COX-2-mediated mechanism in collecting duct cells [31]. There are, however, conflicting opinions regarding the involvement of the PRR system in the hyperactivation of the local RAS in the kidney [9,10,11,12], which plays a role in renal damage associated with diabetes and metabolic syndrome [32,33]. Nevertheless, other homeostatic systems affecting intrarenal RAS activity, such as the copeptin-vasopressin axis, may contribute to the elevation of renin levels, thereby promoting the progression of renal damage, especially in diabetes [34].

Given the design of the present study, which was primarily aimed at assessing changes in expression and localization of FL-(P)RR in different experimental models, our data cannot establish a causal role of (P)RR in renal damage in any of the investigated experimental models. However, overall, the obtained data provide new and valuable insights that should be taken into consideration when investigating the role of this receptor in renal pathologies, especially those associated with cardiometabolic diseases, as well as in renal damage associated with aging or spontaneous forms of FSGS.

Firstly, in MNS rats, these alterations, namely the upregulation of FL-(P)RR, total renin, and COX-2 protein, precede tissue damage, implying their involvement in the pathogenesis of renal injury. Notably, in this genetic model of FSGS, the progression of kidney disease has been linked to elevated glomerular production of thromboxane A2, a major product of COX enzymes derived from arachidonic acid [20,35].

Secondly, the observation that renin immunostaining is evident in tubular and glomerular compartments only after pretreatment of tissue sections with trypsin indicates that a significant portion of the enzyme exists in its precursor form, namely prorenin. Furthermore, the co-localization of total renin with FL-(P)RR immunostaining, along with the upregulation of COX-2 in glomerular and tubular cells, suggests that prorenin undergoes non-proteolytic activation through receptor binding. In any case, it can be inferred that (P)RR receptor signaling is triggered by the binding of its ligands.

Thirdly, while inconsistent with the observed increase in immunostaining, the downregulation of (P)RR gene expression in Diab and HFD-fed mice is consistent with the demonstration of a negative feedback loop, wherein elevated (pro)renin levels suppress (P)RR expression [36]. On one hand, this suggests that the upregulation of gene expression of (P)RR in MNS rats may indicate a disruption in the negative feedback loop in the signaling of (P)RR, serving as a critical factor in the development of renal injury in this genetic model of FSGS. On the other hand, it raises the question of why and how the FL-(P)RR protein is upregulated in the kidneys of Diab and HFD-fed mice. This inquiry prompted us to delve into and investigate potential alterations in the intracellular processing of (P)RR. Previous studies conducted on podocytes have shown that during transit to the plasma membrane, the majority of the FL-(P)RR undergoes cleavage via sequential processing by S1P and the proconvertase furin [24,37]. We found no differences in the expression of S1P and furin. However, in Diab/HFD mice, an increase in the gene expression of Mint3 was observed, contrasting with MNS rats, where no such elevation was identified. Mint3 functions as an adapter protein involved in the signaling and trafficking of membrane proteins. Notably, Mint3 was demonstrated to bind furin and enhance its localization in the trans-Golgi network. Furthermore, the downregulation of Mint3 expression led to elevated furin activity on the cell surface and an enhanced distribution within endosomes [26].

The fourth valuable insight is derived from our in vitro data. As observed in the kidneys of Diab and HFD mice, Mint3 was also upregulated in podocytes cultured under HG conditions, and this upregulation was associated with an increase in cellular FL-(P)RR. It has previously been shown that, under physiological conditions, the majority of FL-(P)RR is primarily situated intracellularly within the Golgi apparatus and endoplasmic reticulum in podocytes. Of this, only a small fraction retains the intact transmembrane and cytoplasmic regions necessary for localization in the membrane, where it is present in a dimeric, non-covalently bound form [24,37]. The majority of FL-(P)RR undergoes cleavage to produce s(P)RR during transit to the membrane, and this cleaved form is subsequently secreted. Additionally, a transmembrane peptide structurally and functionally related to the V-ATPase subunit M8.9, coded by the ATP6ap2 gene, is formed [24,37]. Our observations suggest that hyperglycemia may disrupt the intracellular processing of FL-(P)RR by modulating the gene expression of Mint3, thereby favoring the FL form of the receptor. Of greater significance is the finding that these molecular changes induced by HG were also accompanied by a decreased secretion of s(P)RR by podocytes. This aligns with previous findings indicating significantly lower levels of s(P)RR in diabetic patients compared to nondiabetic patients with chronic kidney disease [38]. Moreover, in a mouse model of metabolic syndrome similar to the one examined in our study, the administration of s(P)RR demonstrated beneficial effects on hyperglycemia, insulin resistance, steatosis, and renal complications [39]. The role of MInt3 in DN has not been studied yet. This adapter protein is primarily investigated in cancer, where it is recognized for enhancing aerobic ATP production through the stabilization of hypoxia-inducible factor (HIF)-1α [40,41]. Notably, it has been demonstrated that, apart from hypoxia, elevated glucose levels also induce HIF-1α activity in inflammatory, vascular, and renal cells. This phenomenon has been linked to the pathogenesis of diabetic complications, including DN [42,43,44,45]. There is no available data on the impact of Mint3 on S1P.

In addition to the aforementioned limitation related to the experimental design that prevents inferring causal relationships, other limitations include the absence of information regarding circulating parameters associated with RAS activity. Furthermore, there is a lack of investigation into the PRR signaling pathways implicated in renal damage and a failure to conduct an in-depth exploration into the molecular mechanisms involved in the regulation of FL-PRR protein levels. Nevertheless, several molecular pathways have been implicated in the damage associated with (P)RR activation [3,28,29,30,31], and the molecular mechanisms that regulate renal cellular levels of FL-(P)RR appear to be specific to the pathogenic models of renal damage. These mechanisms deserve dedicated investigation, which goes beyond the scope of this study.

## 4. Materials and Methods

### 4.1. Immunohistochemical and Morphometric Analysis of Renal Sections

Sections of 4 µm thickness were subjected to microwave treatment (five cycles of 3 min each at 900 W) in citrate buffer pH 6.0. Endogenous peroxidase activity was quenched by incubating sections in 0.3% H2O2 in PBS for 5 min. To prevent nonspecific binding, blocking was performed using Protein Block, Serum-Free (Agilent Dako, Carpinteria, CA, USA). For the detection of (P)RR in mice specimens, sections were incubated overnight at 4 °C with a rabbit polyclonal anti-ATP6AP2 antibody (Sigma-Aldrich, St. Louis, MO, USA, #HPA003156, 1:75), an antibody from the Prestige Antibodies^®^ series developed and extensively validated by the Human Protein Atlas. In the case of rat specimens, the Anti-Renin receptor antibody #ab64957 (Abcam, Cambridge, UK, 1:100) was utilized. Both antibodies recognize the full length (FL) form of the receptor (FL-(P)RR). However, while the #HPA003156 antibody demonstrated excellent results in detecting FL-(P)RR in both mouse tissue specimens and human podocytes (see below), the performance of the #ab64957 antibody surpassed that of #HPA003156 when employed for rat tissue specimens.

Immunohistochemistry (IHC) for mature renin in mouse and rat specimens was performed by incubating sections overnight at 4 °C with a Renin Polyclonal Antibody (Thermo Fisher Scientific, Waltham, MA, USA, #PA5-102432) at a concentration of 5 μg/mL. Two sequential sections of each mouse specimen were stained: one without pretreatment with trypsin to detect only mature renin, and the other after proteolytic trypsin activation of prorenin by incubating sections for 20 min on ice with 0.1% trypsin, to detect total renin (mature renin plus prorenin). Another sequential section was stained for FL-(P)RR to analyze co-localization with total renin. For the detection of COX-2, mouse and rat sections were incubated overnight at 4 °C with a rabbit polyclonal anti-COX-2 antibody (Novus Biologicals, Centennial, CO, USA, #NB 100-689, 1:100).

For all three antigens, the primary antibody incubation was followed by incubation with the corresponding secondary biotinylated goat anti-rabbit IgG (Agilent Dako, #E0432, 1:400) for 1 h at RT. Specificity was confirmed by substituting the primary antibodies with non-immune serum. Positive staining was evaluated in 20 randomly chosen fields of the renal cortex at a final magnification of 400X for mouse specimens and 250X for rat specimens, utilizing the interactive image analyzer Image-Pro Premier 9.2 (Immagini&Computer, Milan, Italy). The results were presented as the mean percentage of the area occupied by the specific stain in these fields. Finally, to facilitate the topographical identification of the glomerular cell type positive for FL-(P)RR, some sections were counterstained with periodic acid-Schiff (PAS) to highlight basement membranes and the mesangial area.

### 4.2. RT-PCR

The mRNA levels of (P)RR, furin, site-1 protease (S1P), and Munc18-1 interacting protein 3 (Mint 3) were evaluated through RT-PCR employing TaqMan gene expression assays (Applied Biosystems, Carlsbad, CA, USA, #4331182; see Table 2).

The purification of total RNA from FFPE tissues was carried out utilizing the RNeasy FFPE Kit (Qiagen, Milan, Italy), following the manufacturer’s instructions. Subsequently, reverse transcription was performed using the High Capacity cDNA Reverse Transcription kit (Thermo Fisher Scientific). The StepOne™ Real-Time PCR System (Thermo Fisher Scientific) was utilized for quantifying the relative gene expression levels and analyzing the data. Gene expressions were determined using the ΔΔCt method and normalized to the control (β-actin expression).

### 4.3. Cell Culture and Treatment

The Human kidney-derived podocyte cell line PODO/TERT256 (Evercyte GmbH, Vienna, Austria, CHT-033-0256) was utilized in this study. PODO/TERT256 cells can be cultured extensively without limitations, all while preserving the physiological characteristics of primary cells. This encompasses sustained expression of cell type-specific markers and functions. The cells were cultured at 37 °C in a 95% air–5% CO_2_ environment using 100 mm^2^ cell culture dishes pre-coated with human collagen I. They were fed with the recommended ready-to-use medium, PodoUp3 (Evercyte, MHT-033-3), supplemented with 10% fetal bovine serum, penicillin (100 U/mL), and streptomycin (100 mg/mL). Cells were grown under normal glucose conditions (NG, 5.5 mM) or high glucose conditions (HG, 20 mM) for 72 h. To assess mycoplasma contamination in the cell cultures, Real-Time (RT)-PCR was performed fortnightly using the MycoSPY Kit (Biontex, München, Germany).

### 4.4. Immunofluorescence Staining for FL-(P)RR in Cultured Human Podocytes

Protein expression and cell distribution of FL-(P)RR in cultured podocytes were assessed through immunofluorescence (IF) analysis. Cells were fixed in 4% paraformaldehyde. Subsequently, a subset of cells for each condition underwent pretreatment with triton X-100 to enhance antibody penetration, while another subset remained untreated to preserve cell membrane integrity. Following this, cells were blocked with 10% normal goat serum for 30 min at room temperature (RT). The cells were then incubated overnight at 4 °C with the #HPA003156 antibody (Sigma, 1:75). This was followed by a 1 h incubation at RT with a fluorochrome-conjugated secondary antibody (Goat anti-rabbit IgG AlexaFluor Plus 488, Thermo Fisher Scientific, #A32731, 1:500). Nuclei were counterstained with Hoechst 33342 (Thermo Fisher Scientific, #3570, 1:2000). IF images were captured using a 40x/0.55 Ph1 objective on a Zeiss Axiovert 200 M fluorescence microscope equipped with an Axiocam 503 color camera, controlled by ZEN 2.0 (blue edition) software (Zeiss, Milan, Italy).

### 4.5. Enzyme-Linked Immunosorbent Assay (ELISA) for s(P)RR

The cellular supernatant was collected subsequent to the treatment of human podocytes with NG or HG. The concentrations of s(P)RR were quantified using the Soluble (Pro)renin Receptor Assay kit from IBL (Immuno-Biological Laboratories Co., Ltd., Minneapolis, MN, USA, #27782), following the manufacturer’s instructions.

### 4.6. Statistical Analysis

The number of biological (i.e., independent experiments) or technical replicates is indicated in figure legends. Results are presented as mean ± SD. Unpaired Student’s *t*-tests with no assumption of equal variance were employed for comparisons between two groups. One-way ANOVA, followed by Tukey’s posttest for multiple comparisons, was used for comparisons involving more than two groups. A *p*-value < 0.05 was considered statistically significant. All statistical analyses were conducted on raw data using GraphPad Prism version 8.00 for Windows (GraphPad Software, San Diego, CA, USA).

## 5. Conclusions

In conclusion, immunostaining for renal FL-(P)RR, (pro)renin, and COX-2 is increased in metabolic and genetic rodent models of renal injury. However, the molecular mechanisms underlying changes in the cellular content of FL-(P)RR are specific to the etiology of renal disease. This observation can serve as a starting point for a more comprehensive exploration into the pathogenic role of the (pro)renin/(P)RR system and drug development for tailored treatments targeting specific renal diseases and cardiometabolic disorders.

## Figures and Tables

**Figure 1 ijms-25-02217-f001:**
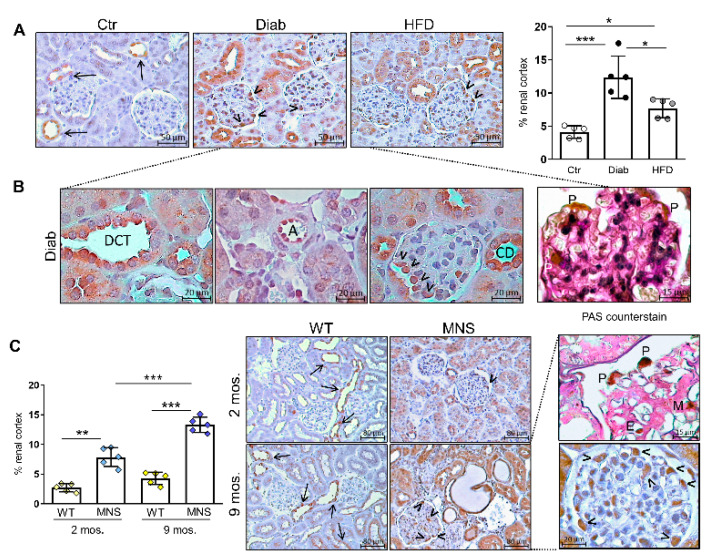
Immunostaining for full length-(pro)renin receptor (FL-(P)RR). Immunohistochemical (IHC) detection of FL-(P)RR and relative quantification of staining in renal specimens from control (Ctr), diabetic (Diab), and high fat diet (HFD) mice (**A**), and characteristics of the staining pattern in kidneys of Diab mice (**B**). IHC detection of FL-PRR, quantification, and immunostaining pattern in Wistar (WS) and Milan normotensive strain (MNS) rats at two and 9 months of age (**C**). Higher magnification images of IHC for FL-PRR in kidneys of Diab mice and 9-month-old MNS rats, counterstained or not with PAS, are shown in panels (**B**) and (**C**), respectively. Bars represent mean ± SD and each dot in (**A**,**C**) represents an individual animal. Arrows indicate positive cells in the tubular compartment, while arrowheads indicate glomerular cells with morphological features and topographical localization characteristic of podocytes. DCT, distal convoluted tubule; A, arteriole; CD, collecting duct; P, podocyte; M, mesangial cell; E, endothelial cell. Post hoc multiple comparison: *** *p* < 0.001, ** *p* < 0.01, * *p* < 0.05.

**Figure 2 ijms-25-02217-f002:**
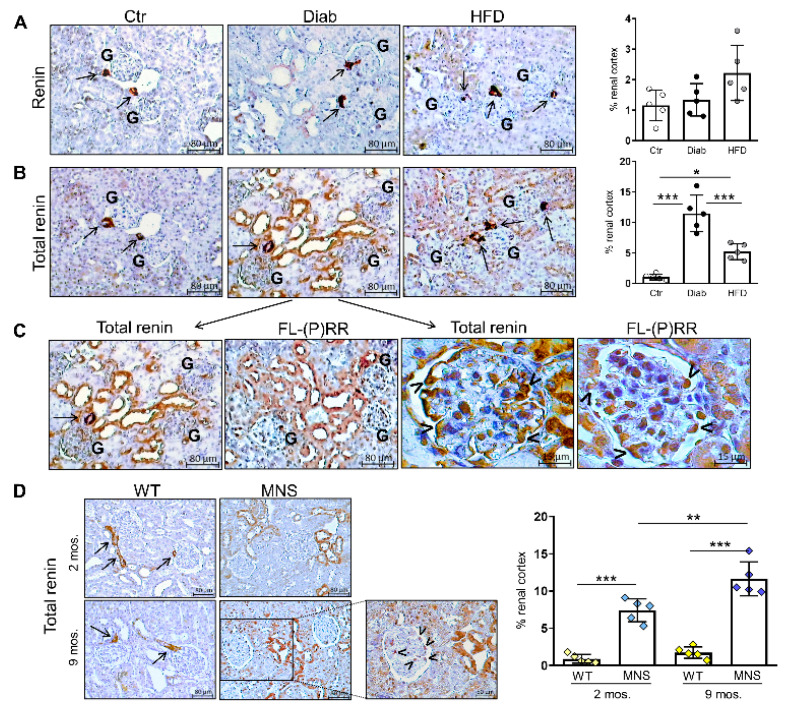
Immunostaining for (pro)renin (total renin). Immunohistochemical (IHC) detection of renin without (**A**) and after (**B**) pretreatment with trypsin in sequential sections, and relative quantification of staining in renal specimens from control (Ctr), diabetic (Diab), and high fat diet (HFD) mice. The IHC detection of full length-(pro)renin receptor (FL-(P)RR) in a sequential section of the same kidney specimens, as depicted in panel B, reveals co-localization with total renin positivity at both tubular and glomerular levels (**C**). IHC detection of total renin and relative quantification of staining in renal specimens of Wistar (WS) and Milan normotensive strain (MNS) rats at two and 9 months of age (**D**). Bars represent mean ± SD and each dot in (**A**,**B**,**D**) represents an individual animal. Arrows indicate positive staining within juxtaglomerular areas, while arrowheads indicate glomerular cells with morphological features and topographical localization characteristic of podocytes. G, glomerulus. Post hoc multiple comparison: *** *p* < 0.001, ** *p* < 0.01, * *p* < 0.05.

**Figure 3 ijms-25-02217-f003:**
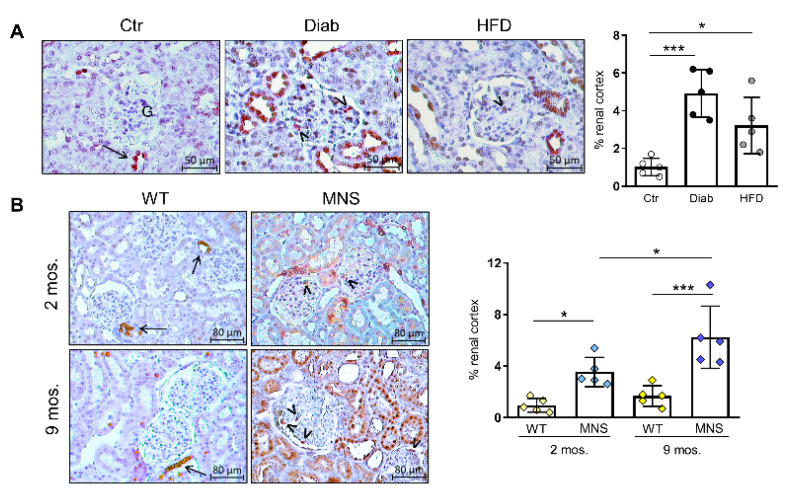
Immunostaining for cyclooxygenase-2 (COX-2). Immunohistochemical detection of COX-2 and relative quantification of staining in renal specimens from control (Ctr), diabetic (Diab), and high fat diet (HFD) mice (**A**), and in Wistar (WS) and Milan normotensive (MNS) rats at two and 9 months of age (**B**). Bars represent mean ± SD and each dot in (**A**,**B**) represents an individual animal. Arrows indicate positive staining of macula densa cells, while arrowheads indicate glomerular cells with morphological features and topographical localization characteristic of podocytes. G, glomerulus. Post hoc multiple comparison: *** *p* < 0.001, * *p* < 0.05.

**Figure 4 ijms-25-02217-f004:**
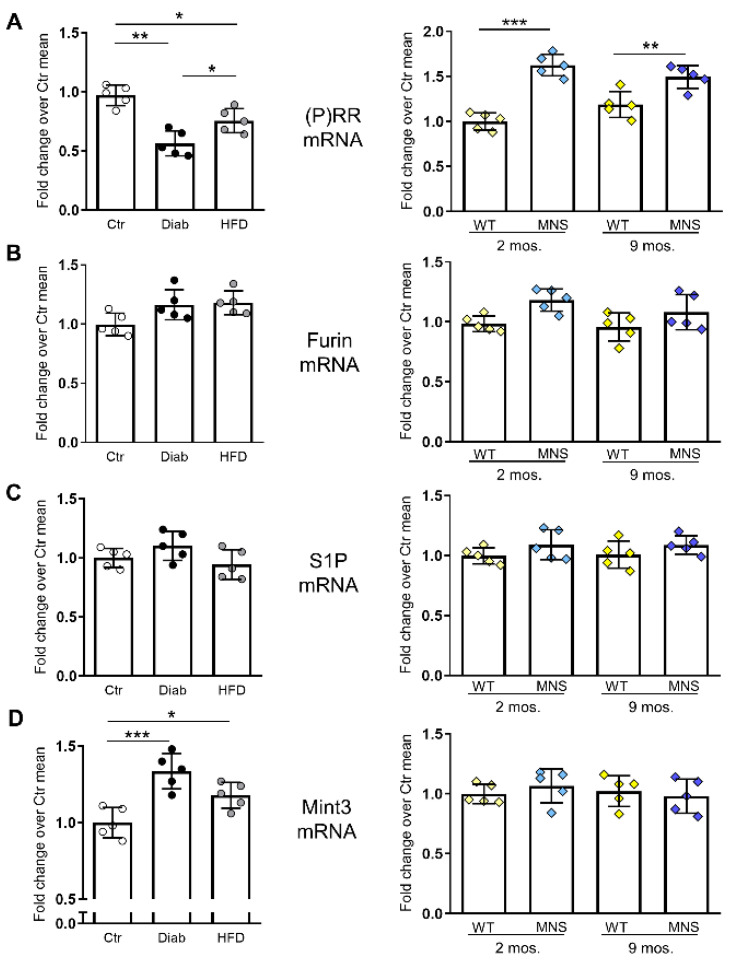
Gene expression of (pro)renin receptor ((P)RR) and associated factors governing its intracellular processing. (P)RR (**A**), furin (**B**), site-1 protease (S1P) (**C**), and Munc18-1 interacting protein 3 (Mint3) (**D**) mRNA levels in kidneys from control (Ctr), diabetic (Diab), and high fat diet (HFD) mice, and in Wistar (WS) and Milan normotensive strain (MNS) rats at 2 and 9 months of age. Bars represent mean ± SD and each dot in (**A**–**D**) represents an individual animal. Post hoc multiple comparison: *** *p* < 0.001, ** *p* < 0.01, * *p* < 0.05.

**Figure 5 ijms-25-02217-f005:**
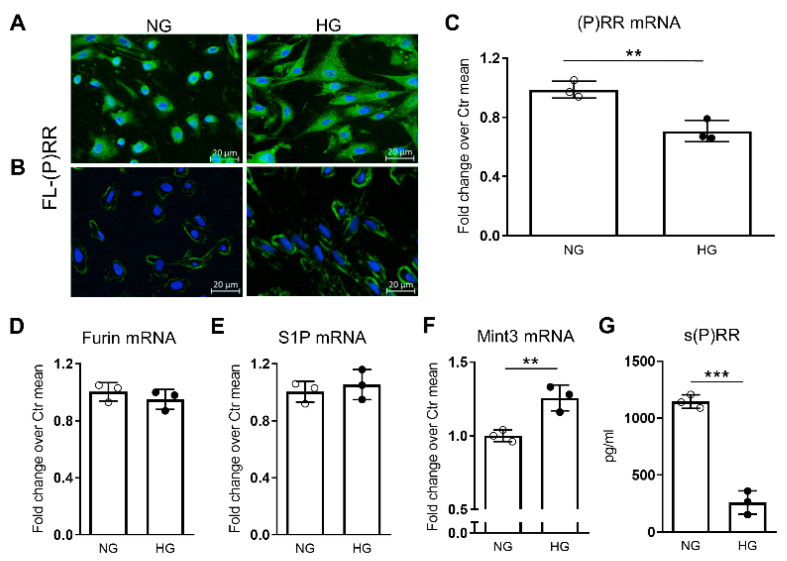
Impact of HG on immunofluorescence (IF) staining and mRNA levels of (pro)renin receptor ((P)RR), gene expression of associated factors governing its intracellular processing, and soluble (s)(P)RR levels in the culture medium of podocytes. IF staining for full length (FL)-(P)RR with (**A**) and without (**B**) pretreatment with triton X-100, and mRNA levels of (P)RR (**C**), furin (**D**), site-1 protease (S1P) (**E**), and Munc18-1 interacting protein 3 (Mint3) (**F**), as well as s(P)RR levels in the culture medium (**G**) of podocytes exposed to high glucose (HG, 20 mM) vs. normal glucose (NG, 5.5 mM) for 72 h. Bars represent the mean ± SD, and each dot represents the mean of two (**C**–**F**) or three (**G**) individual technical replicates for each experimental condition. Post hoc multiple comparison: *** *p* < 0.001, ** *p* < 0.01.

**Table 1 ijms-25-02217-t001:** Metabolic and renal structural and functional features of the three experimental rodent models.

	Diab Mice (4 Months of Diabetes) [16]	HFD-Fed Mice (4 Months of Fatty Diet) [17]	MNS Rats (9 Months Old) [14]
Metabolic phenotype	severe hyperglycemia, insulinopenia (type 1 diabetes), body weight reduction, normal blood pressure	mild hyperglycemia, overweight, hyperinsulinemia, dyslipidemia, normal blood pressure	normal blood glucose, normal body weight, normal blood pressure
Renal lesions	moderate diffuse glomerulosclerosis, increased GSI (↑↑), no tubular damage	mild diffuse glomerulosclerosis, increased GSI (↑), no tubular damage	severe segmental or global glomerulosclerosis, interstitial fibrosis, and tubular damage
Proteinuria	5 times than the control	1.7 times than the control	30 times than the control

Diab = diabetic; HFD = high fat diet; MNS = Milan normotensive strain; GSI = glomerular sclerosis index; ↑↑ = moderate increase; ↑ = mild increase.

**Table 2 ijms-25-02217-t002:** TaqMan Gene Expression assays.

Target		Assay	
	Human	Mouse	Rat
*ATP6AP2*	Hs00997145_m1	Mm00510396_m1	Rn01430719_m1
*Furin*	Hs00159829_m1	Mm00440646_m1	Rn00570970_m1
*MBTPS1*	Hs00921626_m1	Mm00490600_m1	Rn00585707_m1
*Mint3*	Hs01114376_m1	Mm00444450_m1	Rn00582358_m1
*ACTB*	Hs99999903_m1	Mm02619580_m1	Rn00667869_m1

*ATP6AP2* = ATPase H+ transporting accessory protein 2; *MBTPS1* = Membrane-bound transcription site-1 protease; *Mint3* = Munc18-1 interacting protein 3; *ACTB* = Actin Beta.

## Data Availability

The original data used to support the findings of this study are available from the corresponding author [GP] upon reasonable request.

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
