# Peer review of "Renal Expression and Localization of the Receptor for (Pro)renin and Its Ligands in Rodent Models of Diabetes, Metabolic Syndrome, and Age-Dependent Focal and Segmental Glomerulosclerosis"

_ijms, 2024, doi:10.3390/ijms25042217_

Round 1
Reviewer 1 Report
Comments and Suggestions for Authors
The present manuscript by Iacobini and co-workers, describes the expression pattern of (P)RR, renin and COX2 in different animal models of renal damage. The whole study is rather discriptive and lacks insights into the role of (P)RR in the used models or mechanisms contributing to the observed results.
The authors show increased renin positivity in kidney sections. Is the plasma renin activity also altered in the animals? Are Ang II levels in the animals affected? COX2 expression might be increased simply by Ang II stimulation.
Author Response
The present manuscript by Iacobini and co-workers, describes the expression pattern of (P)RR, renin and COX2 in different animal models of renal damage. The whole study is rather descriptive and lacks insights into the role of (P)RR in the used models or mechanisms contributing to the observed results.
We concur with the Reviewer's observation that our study, designed primarily to assess changes in the expression and localization of FL-(P)RR in various experimental models, cannot conclusively establish a mechanistic role of (P)RR in renal damage within any of the investigated experimental models. This limitation was discussed in the original version of the manuscript (see lines 263-265)
The authors show increased renin positivity in kidney sections. Is the plasma renin activity also altered in the animals?
The formalin-fixed and paraffin-embedded kidney samples utilized in this study date back to studies conducted between 2004 and 2013. Unfortunately, we are unable to measure circulating parameters as we have either lost or depleted most of the blood samples during these years. We have now addressed this limitation in the revised version of the manuscript (see lanes 328-329).
However, despite the significantly elevated plasma prorenin levels observed in both diabetic humans [PMID: 7861711] and mice [PMID: 16738017], there appears to be no clear association between diabetes mellitus (DM) and diabetic nephropathy (DN) with plasma renin activity (PRA) [PMID: 32231590; PMID: 16738017]. Consequently, it has been hypothesized that the seemingly low PRA in DN may be misleading, potentially masking an activated intrarenal renin system [PMID: 10541298].
Instead, there is evidence indicating that PRA is associated with body fat accumulation and insulin resistance in obese hypertensive subjects [PMID: 34107965]. However, among the three rodent models investigated, the metabolic syndrome model (mice fed with a high fat diet, HFD) exhibited less renal damage and a less pronounced increase in the components of the PRR system compared to the other two models.
Finally, regarding the rat model of FSGS, we are unaware of any studies investigating the Renin-Angiotensin System (RAS) in the Milan normotensive strain.
Reviewer 2 Report
Comments and Suggestions for Authors
To thank the authors for an original article on Renal expression and localization of the (pro)renin receptor and its ligands in three rodent models of diabetes, metabolic syndrome, and age-dependent FSGN.
It is a well-written manuscript, with an interesting and very clear methodology. It clearly expresses the results of the study and makes a discussion with the most relevant data obtained in it.
There is strong evidence and increasing evidence that the prorenin receptor (PRR) is upregulated in the collecting duct (CD) of the diabetic kidney. PRR activation stimulates fibrotic factors, including fibronectin, collagen, and TGF-β contributing to tubular fibrosis. However, when mentioned about the results of the expression and topographic location of the full length-PRR, no mention is made of it.
I would recommend to make a comment on it, I recommend to add this bibliographical citation:
Gogulamudi, V.R., Arita, D.Y., Bourgeois, C.R.T. et al. High glucose induces trafficking of prorenin receptor and stimulates profibrotic factors in the collecting duct. Sci Rep 11, 13815 (2021). https://doi.org/10.1038/s41598-021-93296-4
Comments on the Quality of English LanguageMinor editing of English language required
Author Response
To thank the authors for an original article on Renal expression and localization of the (pro)renin receptor and its ligands in three rodent models of diabetes, metabolic syndrome, and age-dependent FSGN. It is a well-written manuscript, with an interesting and very clear methodology. It clearly expresses the results of the study and makes a discussion with the most relevant data obtained in it.
We thank the Reviewer for the positive comment to our work.
There is strong evidence and increasing evidence that the prorenin receptor (PRR) is upregulated in the collecting duct (CD) of the diabetic kidney. PRR activation stimulates fibrotic factors, including fibronectin, collagen, and TGF-β contributing to tubular fibrosis. However, when mentioned about the results of the expression and topographic location of the full length-PRR, no mention is made of it. I would recommend to make a comment on it, I recommend to add this bibliographical citation: Gogulamudi, V.R., Arita, D.Y., Bourgeois, C.R.T. et al. High glucose induces trafficking of prorenin receptor and stimulates profibrotic factors in the collecting duct. Sci Rep 11, 13815 (2021). https://doi.org/10.1038/s41598-021-93296-4
We appreciate the suggestion made by the Reviewer and agree with their assessment regarding the importance of (P)RR signaling in activating profibrotic pathways and its role in the pathogenesis of tubular fibrosis. As requested, we have incorporated a comment on this aspect (see lanes 255-257) and included the recommended publication in the revised version of the manuscript (see new Ref. 31).
Reviewer 3 Report
Comments and Suggestions for Authors
The submitted paper analyses the expression of renal FL-(P)RR, (pro)renin and COX-2 in rodent models (healthy controls, streptozocin-induced diabetes, diet-induced metabolic syndrome and age-dependent focal and segmental glomerulosclerosis. Authors provided evidence from different experiments both at the proteomic, genomic and transcriptomic levels. I commend Authors for their nice work trying not just to localise and quantify the expression of FL-(P)RR but also the key players of its regulation and the effectors.
I do not have major comments on the paper. However, I suggest to include in the discussion some of the available literature on hyper activation of intrarenal RAAS system in both diabetes and metabolic syndrome (PMID: 23374893, PMID: 34289001). Also, higher levels of copeptin in diabetes may play a crucial role inducing higher levels of renin (see PMID: 33288413), with important implications in the progression of diabetic nephropathy. Author may consider the impact of other homeostatic systems on intrarenal RAAS activity, such as copeptin-vasopressin axis.
Author Response
The submitted paper analyses the expression of renal FL-(P)RR, (pro)renin and COX-2 in rodent models (healthy controls, streptozocin-induced diabetes, diet-induced metabolic syndrome and age-dependent focal and segmental glomerulosclerosis. Authors provided evidence from different experiments both at the proteomic, genomic and transcriptomic levels. I commend Authors for their nice work trying not just to localise and quantify the expression of FL-(P)RR but also the key players of its regulation and the effectors.
We thank the reviewer for the appreciation of our work.
I do not have major comments on the paper. However, I suggest to include in the discussion some of the available literature on hyper activation of intrarenal RAAS system in both diabetes and metabolic syndrome (PMID: 23374893, PMID: 34289001). Also, higher levels of copeptin in diabetes may play a crucial role inducing higher levels of renin (see PMID: 33288413), with important implications in the progression of diabetic nephropathy. Author may consider the impact of other homeostatic systems on intrarenal RAAS activity, such as copeptin-vasopressin axis.
We appreciate the Reviewer for highlighting this shortcoming. In response to the request, we have incorporated a comment regarding the activation of the intrarenal RAS system in diabetes and metabolic syndrome. Additionally, we have commented on the potential role of other homeostatic systems in regulating local RAS activity (see lanes 257-262). Furthermore, we have included the recommended references (see new Refs. 32-34).
Round 2
Reviewer 1 Report
Comments and Suggestions for Authors
No further comments